# UAV-Based Wireless Data Collection from Underground Sensor Nodes for Precision Agriculture

**Lucas Holtorf** [1]**, Igor Titov** [1]**, Frank Daschner** [2] **and Martina Gerken** [1,*]

[1] Integrated Systems and Photonics, Faculty of Engineering, Kiel University, 24118 Kiel, Germany
[2] Microwave Engineering, Faculty of Engineering, Kiel University, 24118 Kiel, Germany
[*] Correspondence: mge@tf.uni-kiel.de

**Abstract:** In precision agriculture, information technology is used to improve farm management practices. Thereby, productivity can be increased and challenges with overfertilization and water consumption can be addressed. This requires low-power and wireless underground sensor nodes for monitoring the physical, chemical and biological soil parameters at the position of the plant roots. Three ESP32-based nodes with these capabilities have been designed to measure soil moisture and temperature. A system has been developed to collect the measurement data from the sensor nodes with a drone and forward the data to a ground station, using the LoRa transmission standard. In the investigations of the deployed system, an increase in the communication range between the sensor node and the ground station, from 300 m to 1000 m by using a drone, was demonstrated. Further, the decrease in the signal strength with the increasing sensor node depth and flight height of the drone was characterized. The maximum readout distance of 550 m between the sensor node and drone was determined. From this, it was estimated that the system enables the readout of the sensor nodes distributed over an area of 470 hectares. Additionally, analysis showed that the antenna orientation at the sensor node and the drone influenced the signal strength distribution around the node due to the antenna radiation pattern. The reproducibility of the LoRa signal strength measurements was demonstrated to support the validity of the results presented. It is concluded that the system design is suitable for collecting the data of distributed sensor nodes in agriculture.

**Keywords:** precision agriculture; unmanned aerial vehicle (UAV); drone; LoRa; Internet of Things (IoT); Internet of Underground Things (IoUT)

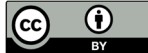

## 1. Introduction

The world's growing population and limited natural resources increase the demand for precision agriculture. The goal is to improve farm management by using information technology in order to increase crop quality and profitability [1]. Additionally, the problems that arise with today's high yield agriculture can also be addressed. Overfertilization results in health hazards, pollution, reduced product quality and a loss of soil fertility [2]. The extensive, uncontrolled irrigation of cropland leads to the deterioration of water sources, salt build-up on fields and soil erosion [3].

The development of precise and cost effective sensing solutions is recognized as one of the key challenges for increasing the adoption of precision agriculture [1]. Monitoring soil parameters, such as moisture, temperature, pH, nitrate or nitrite, is crucial for efficient plant growth and root development [4]. This allows for the precise control of the nutrient content to enhance productivity [5]. Weather, management practices, different soil types and local topology can cause significant variations in the parameters of interest, even within a single agricultural field and over short timespans [6]. Traditional practices, such as *in-situ* measurements conducted by an appropriate operator or offsite laboratory

analysis of soil samples, are not suitable for the spatial and temporal measurement densities required. Thus, the use of distributed, independently operating sensor nodes with *in-situ* measurement capabilities is desired [7].

As the manual operation of wired sensor nodes deployed on large agricultural fields is both impractical and expensive, low-power and long-range data transmission standards, such as Long Range (LoRa), have been established for the transmission of sensor data [8,9]. The sensor nodes should not obstruct the normal work in the field, which introduces the need for wireless underground sensor nodes. Burying sensor nodes in the soil helps to protect them from extreme weather conditions and farm equipment [7]. A drawback is that the achievable communication distance is significantly reduced, because the soil introduces variable high path losses [10]. Researchers at the University of Applied Sciences in Osnabrück have proposed a system for data readout from a sensor buried 60 cm in the ground at a receiving station 350 m away using LoRa communication [8].

Typically, the distances in agriculture are significantly larger than 350 m. Thus, a drone can be a solution for the data collection of sensor nodes distributed in large areas without the need to install additional infrastructure. The deployment of unmanned aerial vehicles (UAVs) in the context of precision agriculture is becoming a valuable tool for farmers and researchers. Barbedo provides a comprehensive review on the use of UAVs for monitoring and assessing plant stresses [11]. Barnetson et al. measured plant pasture biomass and quality with a UAV [12]. Bukowiecki et al. used an UAV with an integrated camera for the estimation of the green area index (GAI) of winter wheat to calibrate the Sentinel-2 data for crop monitoring and yield prediction [13]. Fan et al. discuss plant classification based on drone data [14]. Quino et al. introduced drones in combination with RFID tags for taking a plant inventory [15]. The numerical fluid dynamics simulation by Marturano et al. investigates the air flow around a drone, which is relevant for the placement of sensors on drones [16].

The potential of combing drones with LoRa communication for data collection has recently been proposed in the literature. Behjati et al. demonstrated a system that used drones with an attached LoRaWAN® gateway to collect the data from water quality sensors and livestock monitoring equipment on a farm [17]. The authors investigated the packet loss performance of the LoRa connection, taking into account the different spreading factors and drone speeds. Additionally, they optimized the flight path to maximize the flight range of the fixed-wing vertical takeoff drone. The forwarding of the sensor data from the gateway to a server was conducted using LTE. Park et al. demonstrated a system for collecting environmental sensor data on a tree farm via LoRa using a drone with an attached LoRaWAN® gateway [18]. They successfully applied a proposed client-server communication scheme, but encountered some issues with the gateway in low temperature conditions and limitations of the flight time of the drone due to the additional weight.

Other groups have worked on different subproblems of the approach to collect sensor data with drones. Caruso et al. derived an analytical model for data acquisition with a drone and LoRa from a large regular grid of sensor nodes, as would be found in agriculture [19]. The model allowed for the determination of the optimal spacing of the sensor nodes, the time a sensor needs to be in range of a sensor node and the required velocity of the drone to optimize the probability of successful data collection.

Furthermore, Zhang et al. improved the UAV data collection efficiency of distributed sensor nodes [20]. They developed an algorithm to organize sensor nodes that are close to each other into clusters, with a cluster head that transmits the sensor data of the whole cluster to the drone. A genetic algorithm was then used to minimize the flight path of the UAV between the clusters. Additionally, an adaptive scheme for changing the transmission data rate based on the received signal strength and the signal to noise ratio was reported.

Pan et al. proposed a dynamic UAV speed control scheme in order to adapt to different sensor device densities [20,21]. Based on an analytical model of the connection situation, the number of successful sensor node connections per second was maximized.

Zorbas and O'Flynn worked on the problem of LoRa transmission packet collision when many sensor nodes are located in a specific area. Their protocol allowed for the data collection of 80 nodes in a 1500 m × 1500 m area without packet collisions [22].

In this paper, a system was investigated to collect the data from buried sensor nodes and forward that data to a ground station by using the LoRa transmission standard and a UAV. When operating such a system, a challenge is that it has to be designed in such a way that the data collection is secured while the minimizing battery consumption of the sensor nodes and taking into account the limited flight time of the drone. The necessary optimization of the readout scheme requires knowledge about the different influences on the LoRa transmission quality when the underground sensor node communicates with an aboveground UAV.

To the best of the authors' knowledge, there have not been many published experimental demonstrations on this topic. A similar setup using the NB-IoT standard has been analyzed in simulations by Castellanos et al. The authors demonstrated that an UAV could collect the data within 50 s from 2000 sensors buried in a grid with 10 m spacing under a 20 hectare potato field. Additionally, the impact of soil moisture on the number of successfully served sensor nodes due to changing damping has been shown [23].

Experimental work has recently been published by two other groups. Cariou et al. buried a sensor node at a depth of 15 cm and demonstrated the data collection capability of their system for flight heights of 20 m to 60 m and distances up to 150 m [24]. Hossain et al. developed a similar system and conducted measurements of the received signal strength of the LoRa packages close to the point where the node was buried [25]. A detailed comparison between the system architectures and methods used and the approach described here is given in Section 4.

The general setup (see Figure 1a) consists of a sensor node buried in a field that transmits a series of measurements upon the request of the repeater drone, using the LoRa transmission standard. The repeater drone receives the data and forwards it immediately to the user ground station. Low power sensor nodes were implemented and a protocol for their energy efficient operation was developed. In several experiments, different aspects of the system described were analyzed:

- Dependency of received LoRa signal strength on flight height of the drone and on the burial depth of the sensor node.
- Demonstration of the range extension capability.
- Determination of the maximum distance between the sensor node and drone with the setup.
- Investigation of the effect of the antenna placement inside the sensor node on the received LoRa signal strength.
- Investigation of the repeatability of the signal strength over distance measurements.

This paper is organized as follows. Section 2 presents the materials and methods. Section 3 contains the experimental results. The system design and the results are compared side-by-side with those of the relevant literature in Section 4 (discussion). Finally, the conclusions are presented in Section 5. Some of these results have recently been published as a conference contribution [26].

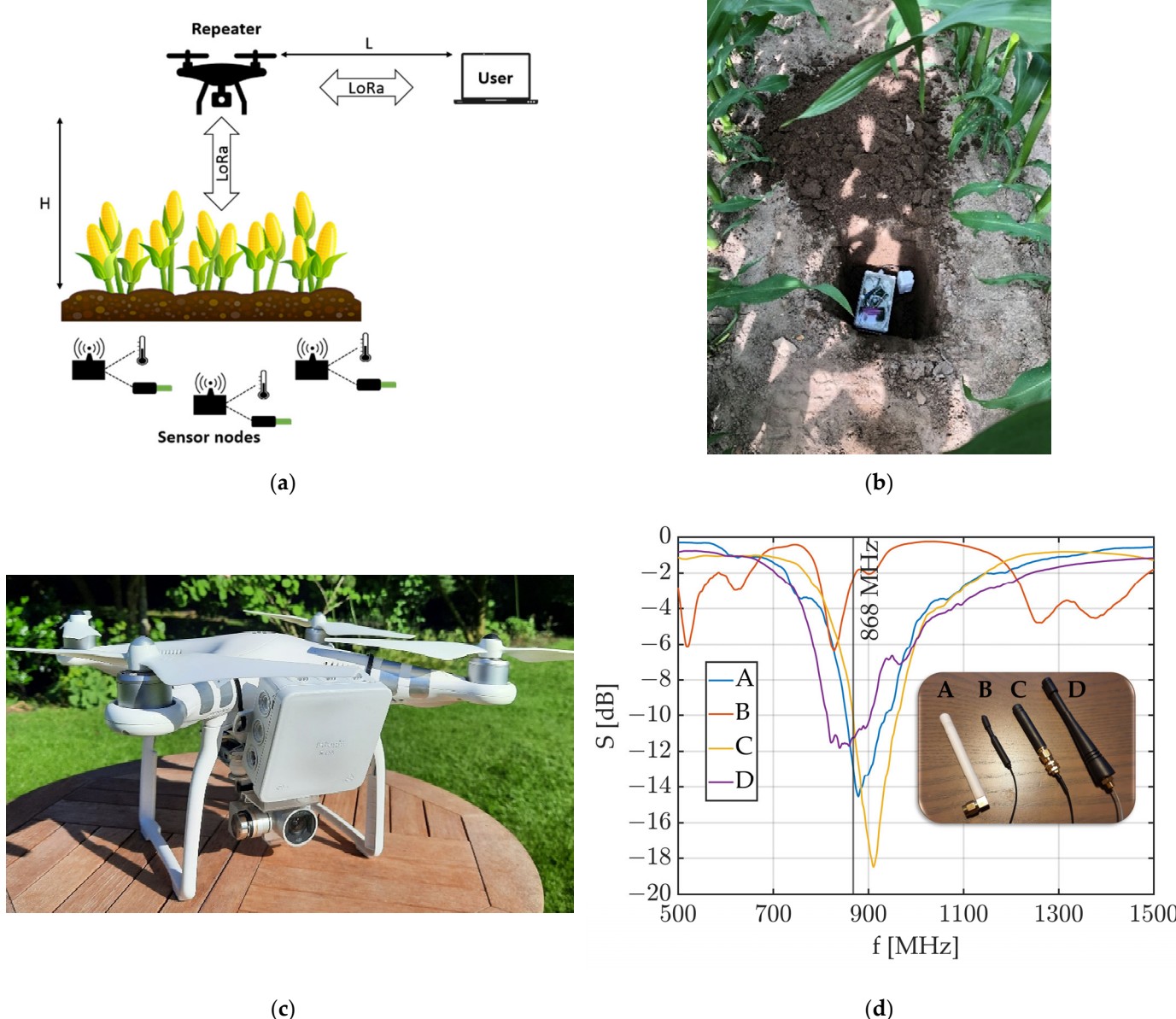

**Figure 1.** (**a**) Schematic of the investigated communication setup. (**b**) Picture of a sensor node deployed in a maize field before closing the hole with soil. (**c**) Picture of the drone with attached component housing. (**d**) Characterization of different antennas in the range around the 868 MHz LoRa frequency with the scattering parameter S.

## 2. Materials and Methods

### 2.1. Materials

Three test sensor nodes, based on the ESP32 boards with a LoRa communication chip operating at 868 MHz, were used (WiFi LoRa 32 V2, Heltec Automation Technology, Chengdu, Sichuan, China). Figure 1d shows the characterization of the different antennas for the LoRa communication. The incorporated antenna from Heltec (Antenna "B") showed an unsatisfactory performance at the desired communication frequency of 868 MHz. For the further experiments, we used the antennas marked as "A" and "C", due to their superior performance around 868 MHz and small antenna size. The spreading factor (SF) was set to 10 to balance between the required power and time required for data transmission and the achievable range. The bandwidth (BW) was set to 125 kHz and the maximum available output power of 20 dBm was used. Those parameters where chosen which

favored longer communication ranges over data rates and transmission power consumption.

A sensor node operates as follows to minimize power consumption. The ESP32 stays in deep sleep mode and wakes up at regular intervals for 5 s to obtain the data from the connected sensors and to listen for data requests from the drone. When it receives a request, it transmits the data stored since the last request and then goes back to sleep. The Heltec board has a rather high deep sleep current, of approximately 1 mA [27]. Therefore, the overall power consumption is determined by the deep sleep power, by the power required while waiting for a request from the drone and by the ratio between the time spent in deep sleep mode vs. time spent waiting for a request. It was decided to favor the transmission range over transmission power consumption because it was assumed that the amount of data collected and the frequency of drone readouts would be low. In the experiments, a wakeup interval of 5 s was chosen to acquire more data about the transmission quality. This interval would need to be increased when operating the system for longer timespans.

Each sensor node is powered by a 3.6 V lithium thionyl chloride (Li/SOCL2) battery (SL-2780, Tadiran Batteries, Kiryat Ekron, Israel), due to the high energy capacity of 19 Ah and long lifetime (up to 10 years). In the experiments, a capacitive moisture and temperature sensor (Adafruit STEMMA Soil Sensor, Adafruit Industries LLC, New York, NY, USA) was connected. An IP68 enclosure protects the components from the ingress of water and dust.

The drone used to investigate the LoRa repeater function was the DJI Phantom 3 Advanced (DJI Technology, Shenzhen, China) (Figure 1c). DJI states a maximum flight time of approximately 23 min [28], but in the experiments, flight times of only approximately 15 min were achieved. A Heltec WiFi LoRa 32 V2 board with antenna "A" was attached to provide LoRa connectivity. It was powered by a 1100 mAh LiPo battery pack. The position of the drone was tracked by the onboard inertial measurement unit (IMU) and an additional NEO-6M GPS module connected to the Heltec board.

In order to acquire the data over LoRa and record it, a ground station consisting of a readout laptop and connected Arduino Uno with a Dragino LoRa Shield (SX1276 Shield-868, Dragino Technology, Shenzhen, China), was used. This module is also based on Semtech's SX1276 LoRa transceiver chip. The received LoRa packages were loaded onto the laptop over a USB serial interface and saved to a CSV file with a Python script. The post-collection analysis of the data was also conducted in Python.

The SX1276 chip provides information about the received signal strength indication (RSSI). This value can be used to evaluate the signal attenuation and was the main parameter used for interpreting the following experiments. Therefore, the RSSI of both LoRa communication paths (sensor to drone and drone to ground station) was acquired. The RSSI of the transmission between the sensor node and the drone was transmitted to the ground station, together with the GPS data from the NEO-6M module and the sensor data.

### 2.2. Methods

In order to analyze the different aspects of the usage of a drone as a repeater for LoRa communication, a series of experiments were conducted on the Experimental Farm Hohenschulen of Kiel University.

### 2.2.1. Investigation of Flight Height and Burial Depth Dependency

In the first series of experiments, the LoRa signal attenuation of the communication between the sensor node and drone was investigated, with the drone hovering at an increasing flight level *H* directly above the sensor node (Figure 2). There were two measurement series with sensor node depth *D* of 30 cm and 60 cm. The RSSI was recorded every 5 m until a 40 m height level was reached.

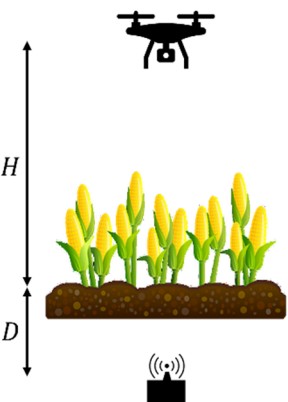

**Figure 2.** Schematic of the experimental setup used for investigating the dependency of the received signal strength at the drone on flight height *H* and burial depth *D*.

### 2.2.2. Demonstration of the Range Extension Capability

In a subsequent experiment, the communication range extension capability of the described system was investigated. Hence, the communication range *L* was compared for two different situations: direct sensor node to ground station communication (S1) and intermediate signal repetition at the drone (S2) (Figure 3).

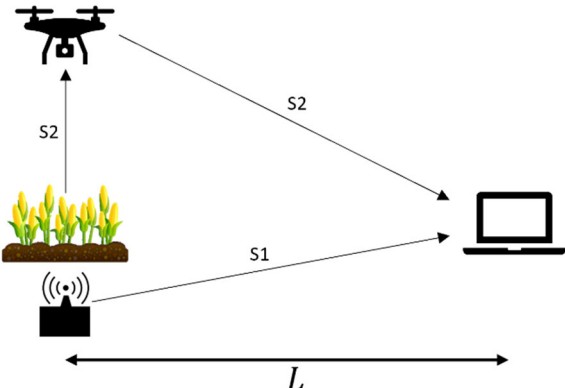

**Figure 3.** Schematic of the experimental setup used for demonstrating the extended range of the data collection system design. S1 shows the direct sensor node to ground station communication path and S2 the path with intermediate signal repetition at the drone.

Firstly, the system was configured so that the sensor node, buried at a depth of 30 cm, communicated directly with the ground station, bypassing the drone. The distance between the sensor node and ground station was increased while measuring the RSSI and the GPS position. Secondly, the experiment was conducted by using the drone as a repeater. The drone hovered 30 m above the sensor node and then, again, recorded the RSSI and GPS position every 50 m while the ground station was moved away.

### 2.2.3. Investigation of the Communication Range between Sensor Node and Drone

When applying the described system in a real-world scenario, a drone would not hover above each sensor node but would conduct the data collection from a certain distance. In order to maximize the area in which the sensor nodes can be deployed and read out, the range *R* for the communication between a sensor node and the drone has to be determined (Figure 4). For this purpose, a sensor node was buried at a depth of 30 cm and the communication was started with the drone hovering above the sensor node. The drone was then flown away at a constant height of 30 m while recording both its GPS position and the RSSI of the packages received at the drone from the sensor node.

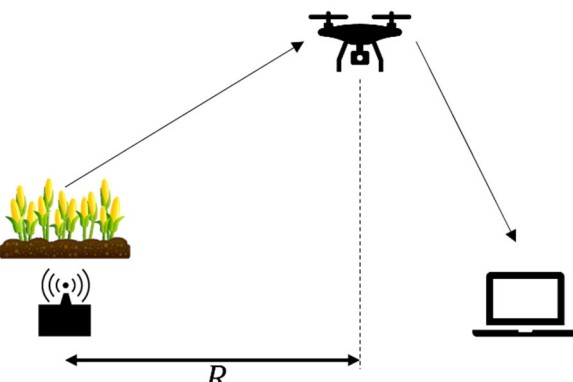

**Figure 4.** Schematic of the experimental setup used for the determination of the communication range between sensor node and drone.

### 2.2.4. Investigation of the Antenna Radiation Pattern

Commercially available antennas for LoRa communication are usually helical antennas. The radiation pattern has a toroidal shape and is symmetric around the z-axis of the antenna (Figure 5b). When using compact enclosures for the sensor nodes, the antennas must be oriented horizontally (Figure 5a). Consequently, the radiation pattern above the ground is not symmetric around the point where the drone is buried.

How this affects the RSSI of the data packages sent from the node to the drone was investigated. To achieve this, the sensor node was buried in a depth of 30 cm, with the antenna z-axis oriented in a known direction in the ground plane. Then, the drone was flown at a constant height of 30 m and with varying positions around the point at which the sensor node was buried. During flight, the sensor data was continuously read out and the GPS coordinates of the drone were acquired each time a package from the sensor node arrived. This enabled the correlation of the RSSI with the position of the drone. The antenna on the drone was oriented with the helix z-axis pointing downwards (Figure 5c) and, therefore, its radiation pattern is symmetric, minimizing distortions caused by the drone orientation.

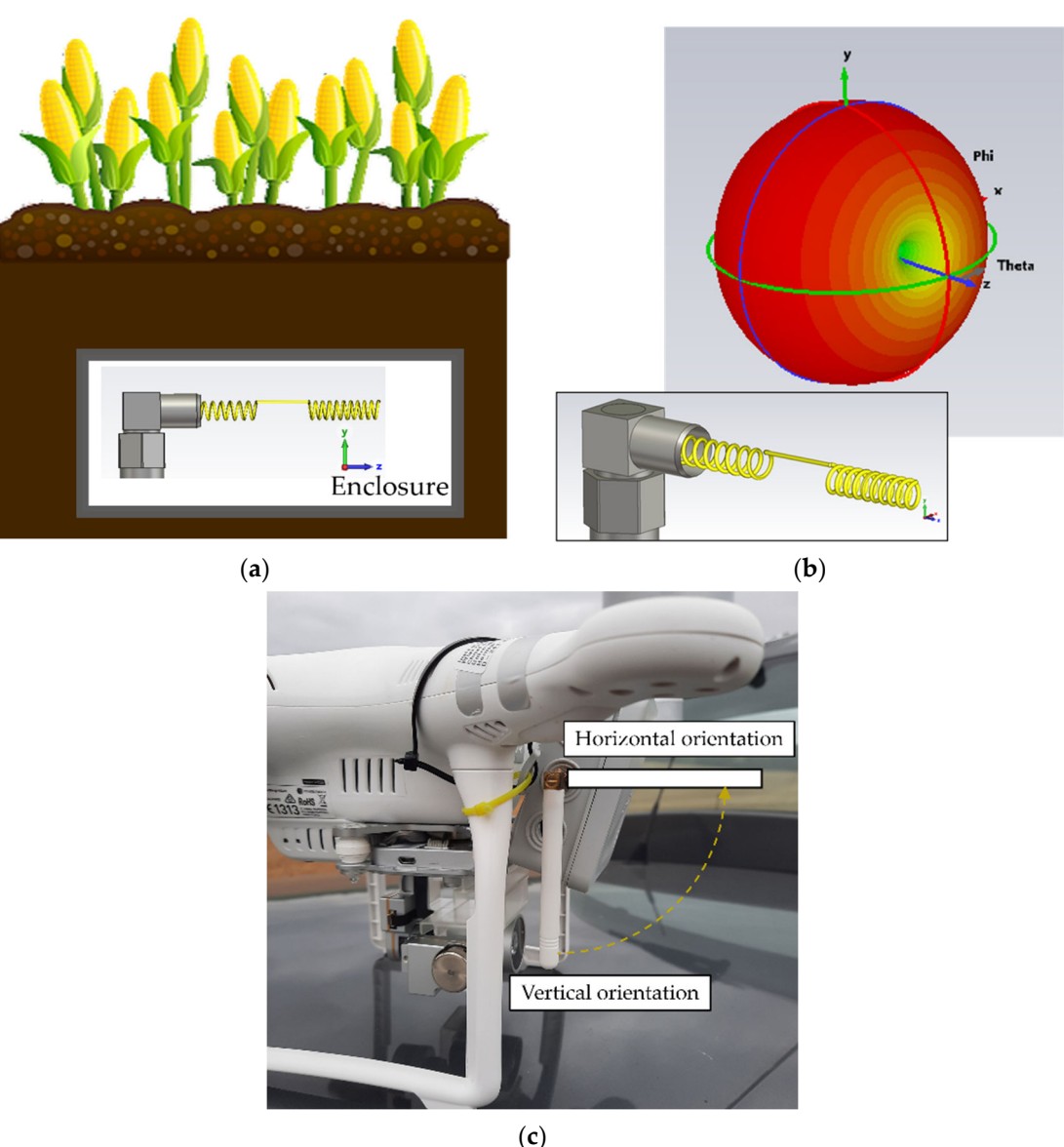

**Figure 5.** (**a**) Schematic of the antenna orientation (z axis) underground in the hardware enclosure. (**b**) Antenna radiation (**c**) Schematic of the antenna orientations investigated.

### 2.2.5. Investigation of Reproducibility

The interpretation of the described experiments requires the reproducibility of the measurement data. In order to verify this, a sensor node was buried at a depth of 30 cm and three flights were carried out with the same path, directly passing above the sensor node at a constant height of 25 m (Figure 6). Each flight was started directly above the sensor node and then the drone was flown away to a defined distance (flight phase P1). Afterwards, it was steered back along the same path, passing over the sensor node, and then flown away again from the sensor node to a turning point (P2). From that point, the drone was flown back to the sensor node (P3). During flight, the data packages were continuously read out from the sensor node. Additionally, the chosen trajectory allows the analysis of how the RSSI changes when flying over the sensor node during data collection.

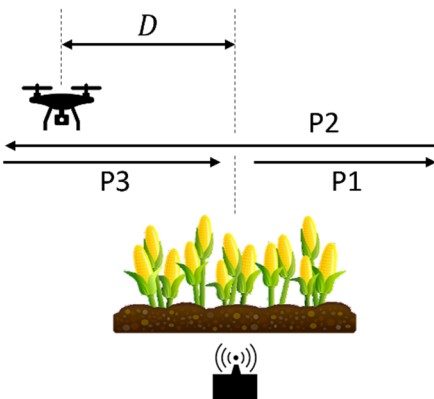

**Figure 6.** Schematic of the experimental setup and the different flight phases.

### 3. Results

*3.1. Investigation of Flight Height and Burial Depth Dependency*

The RSSI for a sensor node depth *D* of 30 cm was −63 dBm at a drone height of 5 m and −77 dBm at a height of 40 m, as shown in Figure 7. For a sensor node depth *D* of 60 cm the values were −64 dBm and −81 dBm, respectively, at drone heights of 5 m and 40 m. During the experiments, a loss of LoRa packages was observed at an RSSI of approximately −120 dBm. From this data, a repeater flight level of 130 m was estimated as possible for a sensor node buried at 60 cm depth. In a second set of experiments, it was found that the attenuation for flying the drone at a constant height to different distances *L* was in a similar range, with ~0.3 dB/m. This experiment was conducted on 23 November 2020.

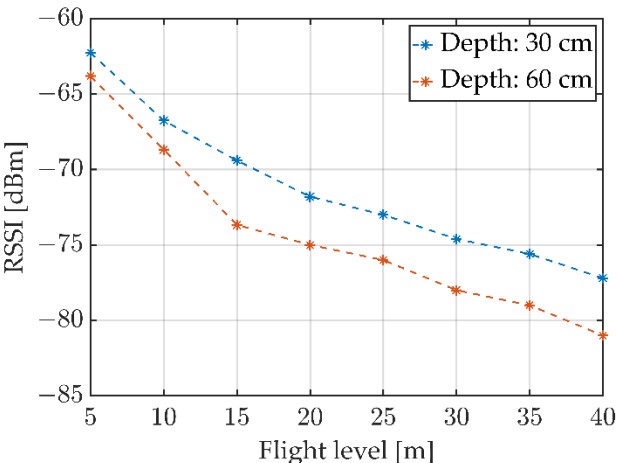

**Figure 7.** Comparison of the relationship between flight level and RSSI for the two sensor node depths of 30 cm and 60 cm.

*3.2. Demonstration of the Range Extension Capability*

Figure 8a shows a satellite image of the paths away from the sensor node for both parts of the experiment. The positions at which the last packages with sensor data were received are marked. The graph depicting the RSSI over distance (Figure 8b) shows that the LoRa packages were received up to *L* = 316 m (blue) for the case without using a drone. When using the drone to receive and transmit the data, the last package was received at a distance of *L* = 1000 m (orange). Thus, the communication range is enhanced by at least three times with the drone as a repeater. The height profile of the path between the sensor node and ground station (Figure 8c) particularly shows the advantage of the system. It can be seen that, for direct communication between node and ground station, the signal is lost after crossing the top of a hill, as the line of sight (LoS) is lost. By using the drone, a

LoS can be ensured even in landscape with hills and other obstacles that might obstruct the signal. This experiment was conducted on 6 August 2021.

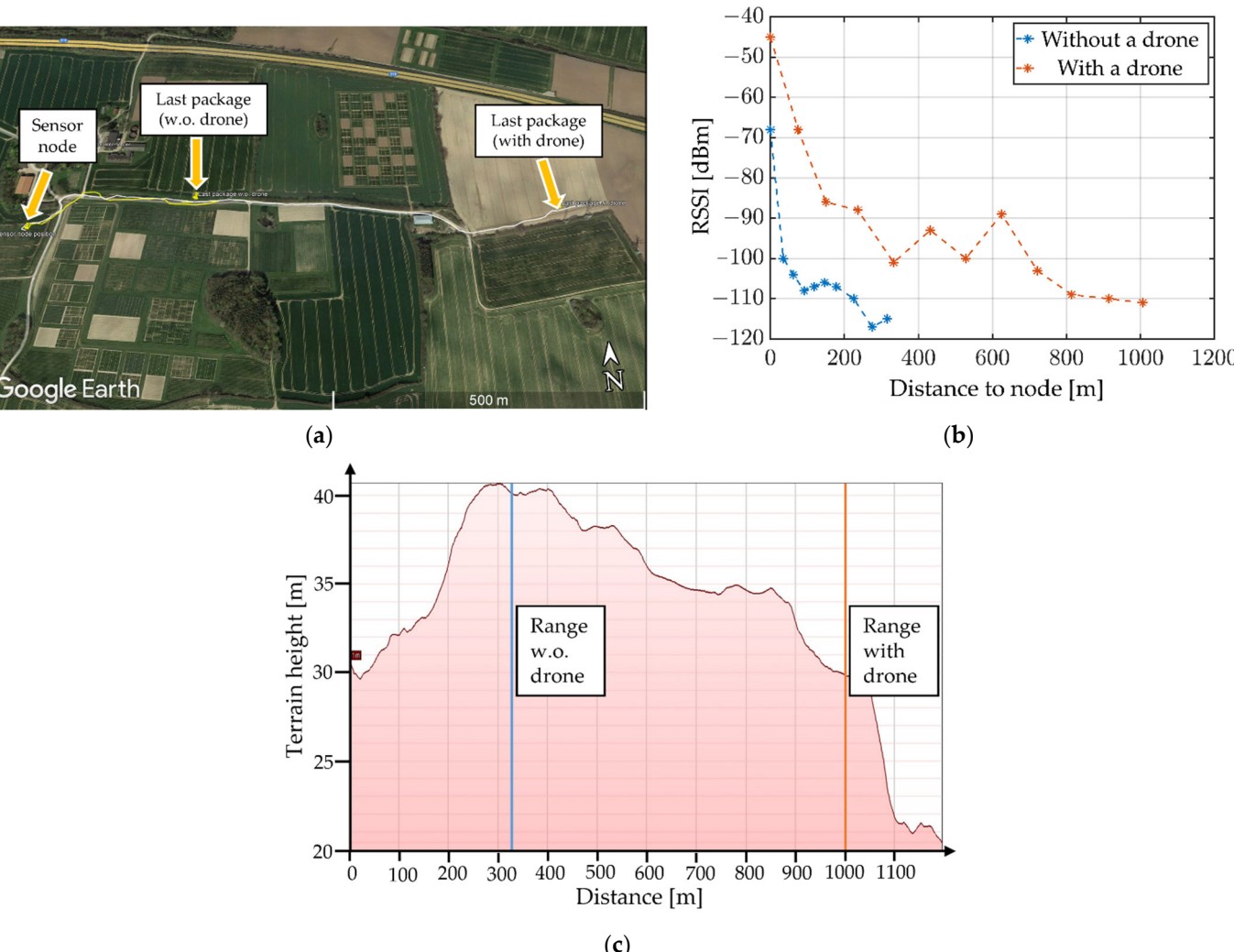

**Figure 8.** (**a**) Satellite view of the travelled path away from the sensor node (Map data: Google, Landsat/Copernicus) (**b**) Comparison of the RSSI of the packages received by the ground station with and without a drone as repeater. Sensor node depth is 30 cm. Drone is hovered 30 m above the sensor node as a repeater. (**c**) Height profile of the traveled path (Map data: Google, Landsat/Copernicus).

### 3.3. Investigation of the Communication Range between Sensor Node and Drone

The RSSI measurement results of the experiment can be seen in Figure 9a. The RSSI decreases with the increasing distance until the last data package is received at a distance of $R = 550$ m. Consequently, the drone can be used as a data collector for sensor nodes within a $R_{max} = 550$ m range.

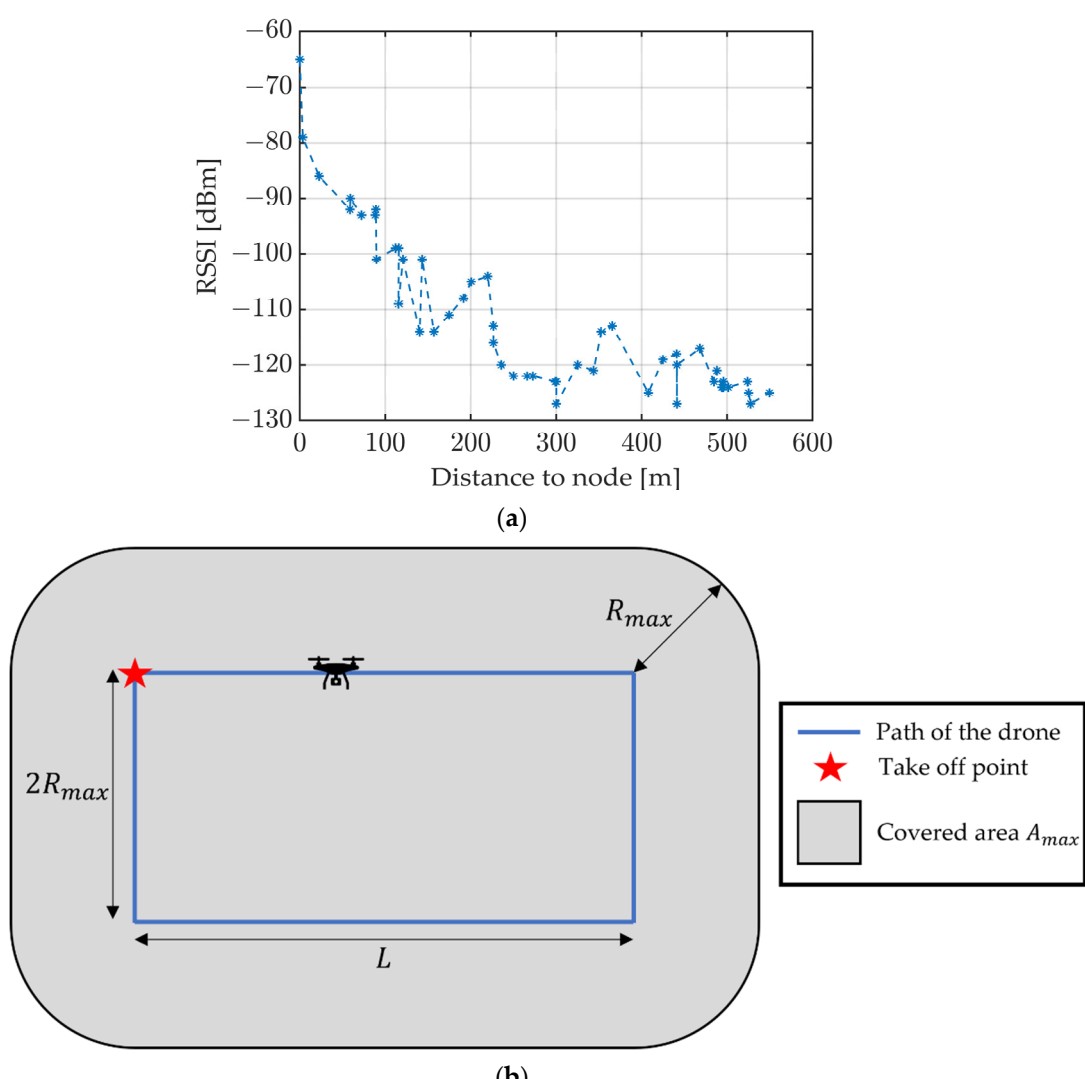

**Figure 9.** (**a**) Reduction in the RSSI with the increasing distance between the sensor node and drone. The sensor node depth was 30 cm. The drone had a constant flight height of 30 m. (**b**) Schematic of the assumed flight configuration for calculating the maximum covered area $A_{max}$ = 470 ha.

From this data, it is possible to estimate the maximum area $A_{max}$ on which the deployed sensor nodes can be read (Figure 9b). Assuming an average flying velocity of 18 km/h and a flight time of 15 min, a maximum flight distance of the drone $S_{max}$ = 4500 m can be calculated. This allows for the determination of the length $L$, considering the given flightpath with minimal overlap (Figure 9b):

$$L = \tfrac{1}{2}(S_{max} - 4R_{max}) = 1150 \text{ m}. \tag{1}$$

Following this, the maximum area

$$A_{max} = 4R_{max}L + (4 + \pi)R_{max}^2 = 4.69 \text{ km}^2 \tag{2}$$

Is calculated. Thus, utilizing a drone for sensor readout enables an area of approximately 470 hectares to be scanned within only 15 min. The data can be transmitted online to the user within a certain range (in this setup approximately 1 km) or by storage and the subsequent transmission of the data packages upon return to the base. This experiment was conducted on the 31 May 2022.

### 3.4. Investigation of the Antenna Radiation Pattern

The results of this experiment are visualized as point clouds (Figure 10). Each colored dot corresponds to the position of the drone when it received a package with the given RSSI. Figure 10a shows the point cloud for the vertical antenna orientation. The RSSI of the packages is higher at the perpendicular point to the antenna orientation than the inline point. This supports the hypothesis that the antenna orientation of the sensor node influences the RSSI of the transmitted packages. This fact can be important when optimizing flight paths for data collection, as the RSSI of data transmission not only depends on the distance between the sensor node and drone, but also on the sensor node orientation.

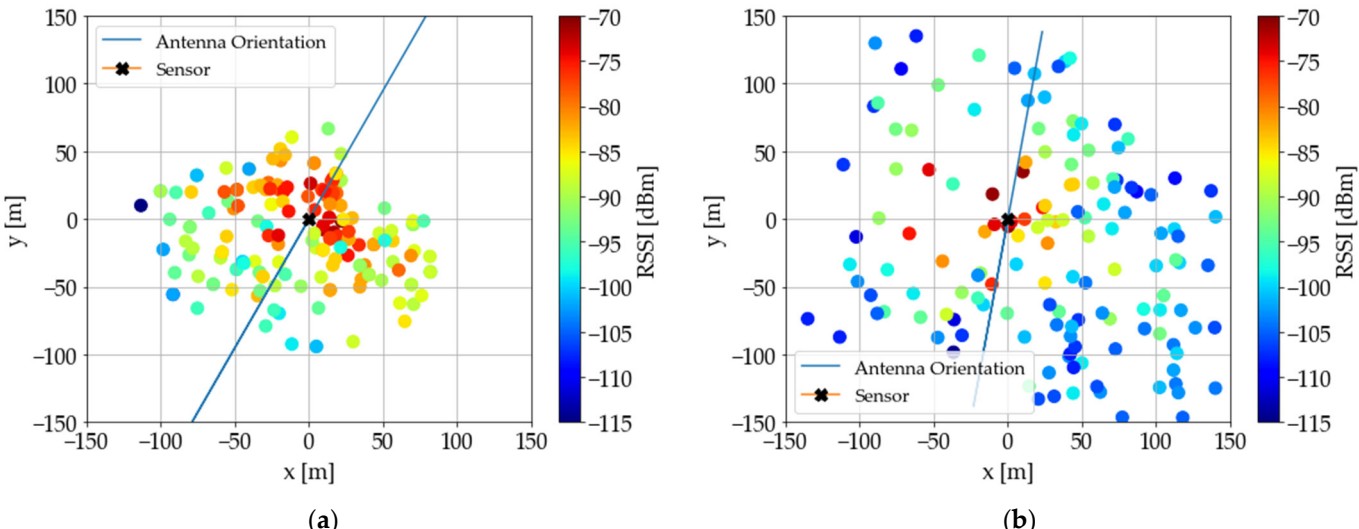

**Figure 10.** (**a**) Visualization of the experimental results for vertical antenna orientation on the drone as a point cloud. Each point corresponds to the position relative to the sensor node that a package was received and is colored to show the RSSI value of the transmission. Additionally, a line is placed along the orientation of the antenna of the sensor node. Sensor node depth was 30 cm. Drone had a constant flight height of 30 m. (**b**) Experimental results for horizontal antenna orientation on the drone.

In addition, the antenna orientation on the drone was changed from a vertical to a horizontal orientation (Figure 10b). This creates an additional dependency of the RSSI on the yaw angle of the drone, as the symmetry of the radiation pattern of the drone's antenna is lost. Therefore, the RSSI is less predictable; therefore, for practical applications, a vertical orientation of the antenna is to be preferred over a horizontal orientation. This experiment was conducted on 7 July 2022.

### 3.5. Investigation of Reproducibility

Firstly, the change in the RSSI over the pathway of a single flight was evaluated (see schematic in Figure 6). When flying away in phase P1, the RSSI decreased with distance D from the sensor node and increased again when turning around to fly phase P2 (Figure 11). After passing over the sensor node, the RSSI decreased, then in P3, it increased again until arriving back over the sensor node. The main observation is that the RSSI is lower when flying away from the sensor node than when flying towards it. It is believed that the reason for this is probably due to the radiation pattern of the antenna. For this experiment, the antenna was mounted with the z-axis facing downwards; thus, the radiation pattern (Figure 5b) is symmetric around the yaw axis of the drone, but not around the pitch axis. When flying towards the sensor node, the node lies closer to the xy-plane of the antenna radiation pattern, where the maximum energy of the incoming signal is received by the drone's antenna. When flying away, the signal from the sensor node comes

from an angle, where less energy of the incoming signal is converted from electromagnetic waves to an electric signal by the antenna, therefore the RSSI is lower.

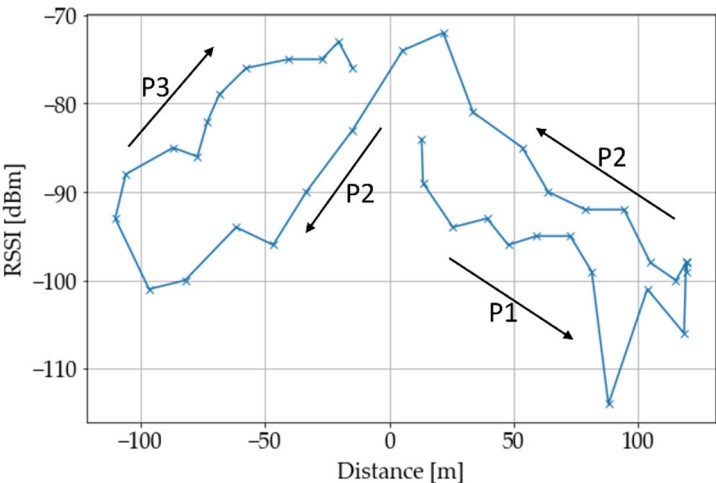

**Figure 11.** RSSI over distance D diagram of the third overflight to explain the behavior.

Secondly, the change in the RSSI over distance was compared between the three flights. During P1, the RSSI values differ, at a maximum of 10 dB for similar distances from the sensor node, and the qualitative development of the RSSI is comparable (Figure 12a). The same is true for the second flight phase (Figure 12b), where the maximum difference is also 10 dBm, but for most distances the difference is smaller, at approximately 5 dB. For the third flight phase, the differences are mostly 4 dB, and when close to the sensor node, the difference goes up to approximately 10 dB (Figure 12c).

The differences can be caused by errors in the reproduction of the desired flight path. These errors are most significant close to the sensor node because, here, small deviations in the flight path can lead to comparatively high deviations in the RSSI value. Additionally, the GPS positioning is also only accurate to a few meters, particularly the NEO6M GPS module as it only uses between six and eight satellites. The RSSI also depends on the pitch angle of the drone, as described before. The pitch angle also determines the speed of the drone. Therefore, speed differences between the flights can cause differences in the RSSI measurements. The large RSSI differences at the turning points of the drone are likely to be explained by this, because the speed and orientation changes at these points are difficult to reproduce between different flights. This experiment was conducted on the 29 August 2022.

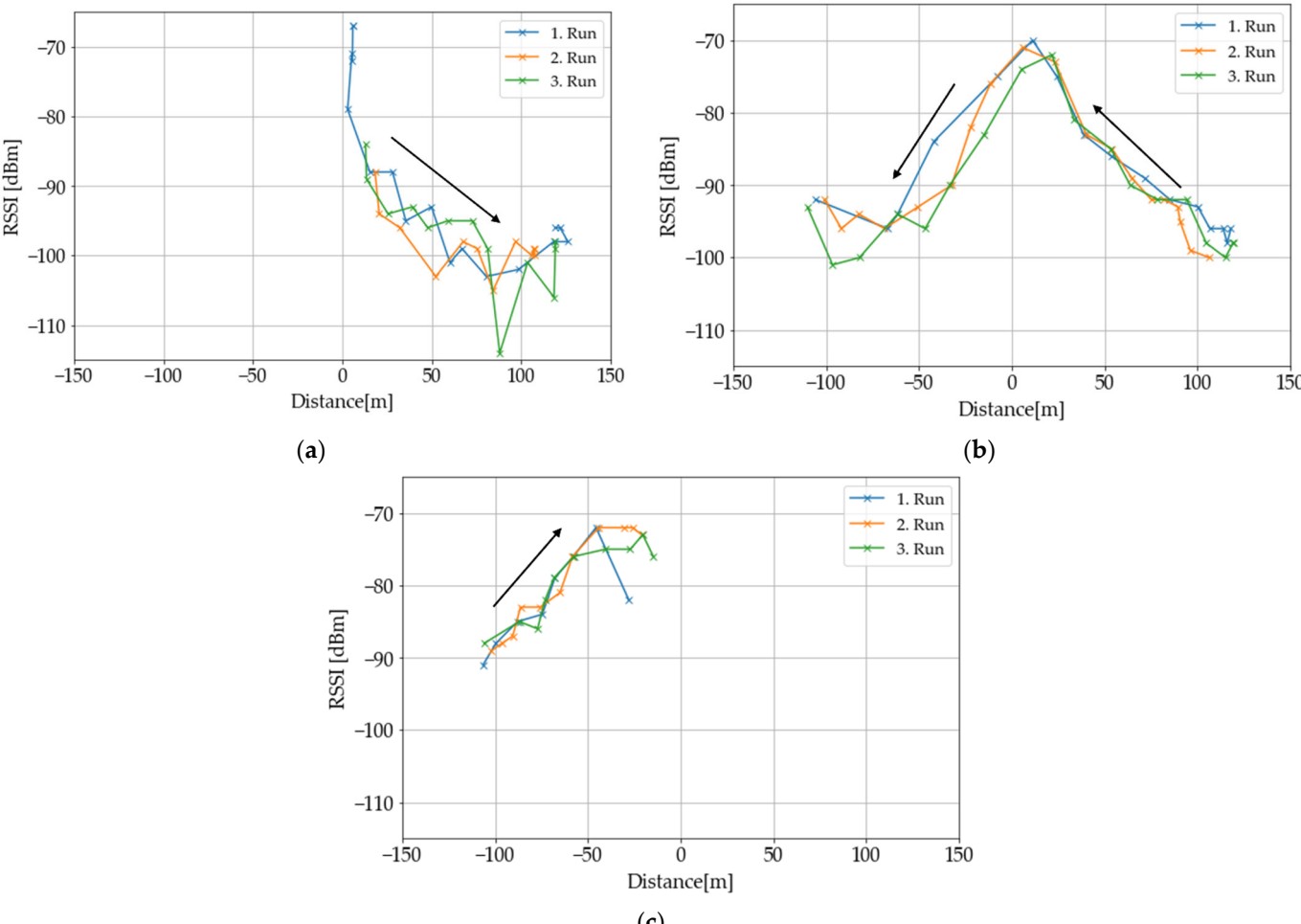

**Figure 12.** RSSI over distance diagram of the three flights, divided into the different flight phases for better clarity. The arrows show the direction of time evolution. Sensor node depth is 30 cm. Drone has a constant flight height of 25 m. (**a**) Phase 1. (**b**) Phase 2. (**c**) Phase 3.

## 4. Discussion

In this section, the system setup, methods and results are compared to the work of Cariou et al. [24] and Hossain et al. [25], because they share the same approach of collecting data from underground sensor nodes by using an UAV and the LoRa transmission technology.

The system setup of Cariou et al. consisted of a sensor node that was designed to be buried in the ground for several months. It awakened regularly from the sleep mode to transmit data from a soil moisture sensor, on request, to a collector node attached to a drone via LoRa. This transmission used an SF of 7, a BW of 125 kHz and a transmission power (TxP) of 14 dBm. The collector node saved the data frames on an SD card and transmitted them over ZigBee to a network gateway when in close proximity.

The setup of Hossain et al. used a sensor node consisting of a Raspberry Pi for control, an RFM95 for LoRa connectivity and a soil moisture sensor. The sensor node was powered from above ground by a 12 V car battery and was not optimized for low power operation. Data packages were sent continuously to a LoRaWAN gateway attached to the drone, configured with SF 7, BW 125 kHz and a TxP of 20 dBm. The gateway also stored the data until the drone landed.

The SF, BW, and TxP for the LoRa transmissions of both groups were configured for higher data rates and lower transmission ranges between sensor node and drone. SF 10, BW 125 kHz, and TxP 20 dBm were used, which has to be considered when comparing the RSSI values and achieved transmission ranges. Another difference was the way the

data from the drone was further processed after being received. Both Cariou et al. and Hossain et al. used an approach in which the data was collected and was only accessible to the user at the end of the flight. The data packages were transmitted online to the user with LoRa. This allowed for the diagnoses of problems during flight and access the data without delay. However, this also reduced the data rate further as the drone LoRa module could not collect new data packages from other nodes while retransmitting old packages to the ground station. Another approach was the direct upload of the data to the cloud via a 4G internet connection, as demonstrated in [17,29]. This obviously requires the local availability of a 4G connection.

In their experiments, Cariou et al. first measured the RSSI in a range of distances (0 to 150 m) from the UAV to the buried sensor node (15 cm deep) and for heights of 20/40/60 m. Their results are comparable to the results in Sections 3.1 and 3.3, which considered a 30 m flight height and 30 cm node depth. Their data showed that for high enough lateral distances, the RSSI is higher for higher flight heights. Secondly, they conducted an over-flight experiment over the buried sensor node at different UAV speeds to show that the system was able to collect 100 packages from the node, independent of the speed. Their results showed RSSI measurements that are comparable to the results in Section 3.5, but the authors did not further investigate or explain this behavior. The authors further optimized the path for the data collection of 25 buried sensor nodes.

Hossain et al. measured the RSSI with a receiver gateway attached to a wooden pole with increasing height (0.3 to 2 m) and increasing lateral distance (0 to 5 m) relative to the sensor node (30 cm node depth), both for cross- and co-polarization between the sensor node and gateway antenna. The results were also compared with numerical simulations. Furthermore, the authors measured the packet loss for distances between the node and drone of up to 82 m, again comparing cross- and co-polarization. Additionally, they conducted trajectory flights at four heights, from 0.91 m to 3.66 m around the sensor node, covering a rectangle spanning −24 to 24 m and −27.4 m to 24 m. During flight, they again measured the RSSI for cross- and co-polarization. This experiment, although methodically similar to Section 2.2.4, considered a smaller lateral range scale and smaller flight heights. From the results, the authors concluded that the RSSI was largely independent of the polarization. The RSSI measurements are mostly complementary to the work of Cariou et al. [24] and the work described here as they focused on a scale closer to the buried sensor node.

Overall, this work shows similarities to both these other works when the methods and the results are considered. A common point of interest is the RSSI distribution around the sensor node, as described before, but different length scales were investigated. Furthermore, an attempt was made to estimate the maximum coverable area for the system setup. A particular new emphasis of this work is the impact of the orientation of the antenna at the sensor node and at the drone. Additionally, experiments on using a LoRa link from drone to ground station are presented.

Comparing the chosen system setups and methods, it can be observed that there are different assumptions about the sensor node density and the overall amount of data that needs to be collected. Cariou et al. and Hossain et al. considered smaller length scales in their experiments and configured the LoRa communication more for higher data rates than for communication distances. This is suitable for higher sensor node densities or if large amounts of data need to be transmitted at each connection. The system described here was designed more towards covering a greater area with lower sensor node density and lower amounts of data for each transmission.

## 5. Conclusions

In conclusion, a setup for communication with an underground network of LoRa sensor nodes has been demonstrated using a drone to collect the data. A much larger area can be covered than previously, without the need for the installation of additional ground-based hardware. The maximum communication ranges between an underground sensor

node and drone and the dependency of the received signal strength on the flight height and burial depth for the setup was experimentally determined. The evaluation of the antenna orientation of the sensor node and the overflight experiment hinted at other possibilities for drone flight path optimization. In addition, the reproducibility of the RSSI measurements for different distances between the sensor node and drone was also demonstrated.

The long-term goal is a fully autonomous data collecting system for several sensor nodes based on a drone. The sensor nodes were operated in the soil and demonstrated successful connection for up to 49 days. Currently, sensors for physical and chemical soil parameters are integrated into the sensor nodes for the relevant data collection. For this purpose, a microfluidic unit for the extraction of soil water has been developed [30], as well as a compact sensor chip based on organic optoelectronic devices for nitrite and nitrate determination [31]. To enable the data collection for a larger number of sensor nodes, a more sophisticated communication protocol must be implemented to avoid package collisions. Additionally, adapting the spreading factor and the bandwidth according to the respective distances and sensor node densities is important to maximize the area coverage and collection performance of the system. This can be further improved by optimizing the drone path between sensor nodes. Regarding the system setup, it is still to be determined whether LoRa is the optimal way to transmit the data from the drone to the ground station. Possibly, a transmission directly over LTE to a cloud is advantageous to point-to-point LoRa, particularly for situations where high data rates are required. Another important aspect is the optimization of the sensor node antenna to the ground-to-air transmission situation.

**Author Contributions:** Conceptualization, L.H., I.T., F.D. and M.G.; methodology, L.H., I.T., F.D. and M.G.; software, L.H.; validation, L.H. and I.T.; formal analysis, L.H., I.T. and M.G.; investigation, L.H. and I.T.; resources, M.G.; data curation, L.H. and I.T.; writing—original draft preparation, L.H., I.T. and M.G.; writing—review and editing, L.H., I.T., F.D. and M.G.; visualization, L.H., I.T. and F.D.; supervision, I.T. and M.G.; project administration, M.G.; funding acquisition, M.G. All authors have read and agreed to the published version of the manuscript.

**Funding:** This project received funding from the European Regional Development Fund (EFRE) by the European Union (OPTOCHIP, LPW-E/1.2.2/1303).

**Data Availability Statement:** Data underlying the results presented in this paper are not publicly available at this time but may be obtained from the authors upon reasonable request.

**Acknowledgments:** The authors thank the Institute of Crop Science and Plant Breeding for providing the drone and for farm access at the Experimental Farm Hohenschulen.

**Conflicts of Interest:** The authors declare no conflict of interest.

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
