# Peer review of "UAV-Based Wireless Data Collection from Underground Sensor Nodes for Precision Agriculture"

_agriengineering, doi:10.3390/agriengineering5010022_

Round 1

Reviewer 1 Report

Dear authors,

I have reviewed your paper on using a drone and LoRa communication to collect data from underground sensor nodes in precision agriculture. Overall, your system appears to be suitable for the described purpose. However, I have some suggestions for improvement:

  1. The literature review in your paper discusses the use of wireless point-of-source systems, specifically Long Range (LoRa) communication, for soil monitoring in precision agriculture. However, it could be more thorough and structured to better serve as a paper introduction on this topic. Consider including more context and background information on the importance of soil monitoring in precision agriculture, and discussing the limitations and challenges of using drones and LoRa communication for data collection. You could also group studies into categories based on the specific focus or methods used to provide more structure and organization.

  2. The material and methods section could be more detailed and specific. Provide more information on LoRa communication parameters and configurations, sensor nodes (including sensors and data collected), the drone (including flight time and payload capacity), and the ground station (including the software used for data acquisition and analysis).

  3. The results section and the methodology are currently mixed up and could be separated and made more coherent. Consider separating the results and methodology into distinct sections, and being clearer in your presentation of the results.

  4. The discussion and conclusion sections are also mixed up and could be separated for clarity. Consider separating the discussion into its own section, and discussing the results side-by-side with the relevant literature.

  5. In all, I do not see the novelty of your paper. It seems to simply be showing a framework of data collection from drones using well-known IoT tools. Please be more precise and show where you stand in the state of the art of field.

Thank you for your efforts in this research. I hope these suggestions are helpful in improving your paper.

Sincerely,

Reviewer 2 Report

Authors have done good work and in my opinion paper is technically sound and it is accepted in current form 

Reviewer 3 Report

See the file

Round 2

Reviewer 1 Report

Dear Authors,

Thank you for your efforts in the paper and for revising it in response to our comments. I have read the paper and can see the corrections clearly. However, I still have concerns about the coherence and clarity of the purpose, particularly in the abstract. I suggest addressing the following points:

1- Start with a brief overview of precision agriculture and its objective.

2- Describe the underground sensor nodes used in the system and the parameters they monitor.

3- Provide more information about the data gathering system, including the use of LoRa connectivity and a drone.

4- Mention specific research outcomes, such as increased communication range, factors influencing signal strength, and expected readout area.

5- Highlight the results of experiments on the impact of antenna designs and the reproducibility of signal strength tests.

6- Conclude with a clear and straightforward statement on the suitability of the system design for its intended purpose.

I am satisfied with the overall presentation of the paper and hope to see the abstract in a clearer format.

Best regards,

Author Response

Dear reviewer,

thank you for the detailed suggestions on how to improve our abstract. We have rewritten the abstract accordingly.

Best regards,
Martina Gerken